# Additive genetic effect of *GCKR*, *G6PC2*, and *SLC30A8* variants on fasting glucose levels and risk of type 2 diabetes

**Guanjie Chen**[1☉], **Daniel Shriner**[1☉], **Jianhua Zhang**[2], **Jie Zhou**[1], **Poorni Adikaram**[3], **Ayo P. Doumatey**[1], **Amy R. Bentley**[1], **Adebowale Adeyemo**[1], **Charles N. Rotimi**[1]*

**1** Center for Research on Genomics and Global Health, National Human Genome Research Institute, Bethesda, Maryland, United States of America, **2** Metabolic Diseases Branch, National Institute of Diabetes and Digestive and Kidney Diseases, Bethesda, Maryland, United States of America, **3** Advanced BioScience Laboratories, Rockville, Maryland, United States of America

☉ These authors contributed equally to this work.
* rotimic@mail.nih.gov

**Data Availability Statement:** The AADM, CADM, and HUFS datasets used and/or analyzed in the current study are available from the corresponding author upon reasonable request. The dbGaP data

## Abstract

Impaired glucose tolerance is a major risk factor for type 2 diabetes (T2D) and several cardiometabolic disorders. To identify genetic loci underlying fasting glucose levels, we conducted an analysis of 9,232 individuals of European ancestry who at enrollment were either nondiabetic or had untreated type 2 diabetes. Multivariable linear mixed models were used to test for associations between fasting glucose and 7.9 million SNPs, with adjustment for age, body mass index (BMI), sex, significant principal components of the genotypes, and cryptic relatedness. Three previously discovered loci were genome-wide significant, with the lead SNPs being rs1260326, a missense variant in *GCKR* ($p = 1.06 \times 10^{-8}$); rs560887, an intronic variant in *G6PC2* ($p = 3.39 \times 10^{-11}$); and rs13266634, a missense variant in *SLC30A8* ($p = 4.28 \times 10^{-10}$). Fine mapping, genome-wide conditional analysis, and functional annotation indicated that the three loci were independently associated with fasting glucose. Each copy of an alternate allele at any of these three SNPs was associated with a reduction of 0.012 mmol/L in fasting glucose levels ($p = 8.0 \times 10^{-28}$), and this association was replicated in trans-ethnic analysis of 14,303 individuals ($p = 2.2 \times 10^{-16}$). The three SNPs were jointly associated with significantly reduced T2D risk, with an odds ratio (95% CI) of 0.93 (0.88, 0.98) per protective allele. Our findings implicate additive effects across pathophysiological pathways involved in type 2 diabetes, including glycolysis, gluconeogenesis, and insulin secretion. Since none of the individuals homozygous for the alternate alleles at all three loci has T2D, it might be possible to use a genetic predictor of fasting glucose levels to identify individuals at low *vs.* high risk of developing type 2 diabetes.

## Introduction

Impaired fasting glucose, also referred to as prediabetes, is a risk factor for cardiovascular disease and type 2 diabetes (T2D) [1, 2]. Investigating the genetic architecture of fasting glucose

are deposited in dbGaP and available through dbGaP authorized approval.

**Funding:** This research was supported by the Intramural Research Program of the Center for Research on Genomics and Global Health (CRGGH). The CRGGH is supported by the National Human Genome Research Institute, the National Institute of Diabetes and Digestive and Kidney Diseases, the Center for Information Technology, and the Office of the Director at the National Institutes of Health (grant 1ZIAHG200362 to CNR). The funders had no role in study design, data collection and analysis, decision to publish, or preparation of the manuscript.

**Competing interests:** The authors have declared that no competing interests exist.

will lead to a better understanding of the mechanisms involved in glucose homeostasis and subsequently the pathophysiology of T2D [3]. Genetic analysis of fasting glucose as a quantitative trait complements genetic analysis of T2D as a dichotomous trait.

Genome-wide association studies (GWAS) have been widely used in investigating the genetic architecture of fasting glucose levels. Genetic associations with fasting glucose have been reported in 17 loci in individuals of European ancestry [3–5]. There are more than 240 published loci associated with T2D [6, 7]. Only nine T2D loci (*GCKR*, *GCK*, *SLC30A8*, *PROX1*, *ADCY5*, *DGKB*, *GLIS3*, *TCF7L2*, and *MTNR1B*) overlap with fasting glucose loci, which appear to mediate impairment of the glucose-sensing machinery in pancreatic β islet cells [3]. One trivial explanation is low power. Alternatively, loci affecting physiological levels of fasting glucose among normoglycemic individuals need not be the same as loci that affect pathophysiological levels of fasting glucose when hyperglycemic individuals are also considered. As the genetic architectures of fasting glucose and T2D are incompletely known, we caution against overinterpreting this interim result.

The Atherosclerosis Risk in Communities (ARIC) study is a prospective study of atherosclerosis in middle-aged adults [8]. Previously, a GWAS for the average of four fasting glucose measurements taken over nine years was conducted in individuals without prevalent diabetes, and three known loci near *MTNR1B* (rs10830963), *GCK* (rs2971669), and *G6PC2* (rs853787) were replicated [4]. Here, we defined the outcome as the first fasting glucose measurement from all untreated individuals, *i.e.*, non-diabetic individuals as well as untreated diabetic individuals. We then performed a GWAS using a linear mixed model with a high-density imputation reference panel and identified three associations in loci previously reported to influence fasting glucose (*GCKR*, *G6PC2*, and *SLC30A8*). Associations at two missense variants in *GCKR* (rs1260326) and *SLC30A8* (rs13266634) were identified in individuals with European ancestry and all three associations replicated in trans-ethnic meta-analysis. These three associations also affect risk of T2D, indicating not just physiological relevance to fasting glucose levels but also pathophysiological relevance to T2D.

## Materials and methods

The Atherosclerosis Risk in Communities study is a prospective study of clinical atherosclerotic diseases [8]. Individual-level genotype and phenotype data were obtained by authorized access to dbGaP (https://www.ncbi.nlm.nih.gov/gap/). T2D case status was defined as fasting glucose ≥7.0 mmol/L, self-report of a diagnosis by a physician, or current diabetic treatment. For fasting glucose analysis, individuals without T2D (8,902) and with untreated T2D (330) were used; individuals without diabetic treatment were included because their fasting glucose values were unaffected by treatment. The inclusion of untreated cases makes our analysis more powerful than previous analysis of normoglycemic individuals. Selected variables included age, sex, body mass index (BMI), fasting glucose, and T2D status. Among individuals with a reported race of White, a total of 9,232 individuals without T2D or with untreated T2D were included and used for analysis of fasting glucose. Similarly, a total of 9,731 individuals were used for analysis of T2D.

Fasting serum samples were assayed for glucose and were measured on the Roche Hitachi 911 analyzer using the hexokinase method (Roche Diagnostics). Age, sex, race, and ethnicity were self-reported. BMI was calculated as body weight (in kilograms) divided by height (in meters) squared. Medication history over a period of two weeks prior to the visit was verified by review of medication containers that participants brought to the visit.

## Genotyping and imputation

Genotyping was performed on the Affymetrix Genome-wide Human SNP Array 6.0. After quality control for minor allele frequency (MAF) $\geq 0.01$, genotype call rate $\geq 0.95$, per-individual missingness rate $\leq 0.05$, and a Hardy-Weinberg equilibrium test $p$-value $> 10^{-6}$, we retained 800,099 autosomal SNPs. Imputation was performed using the Sanger Imputation Service (https://imputation.sanger.ac.uk/) with the IMPUTE2 software [9] and the 1000 Genomes Project Phase 3 reference panel [10]. The resulting imputed SNPs were filtered for MAF $\geq 0.01$ and info score $\geq 0.7$ [11]. After filtering, 7,896,808 SNPs were retained for association analysis. Coordinates were based on the hg19 build. All alleles are reported with respect to the positive strand.

## Association analysis

Fasting glucose levels from the first available measurement were included (S1 Fig). Association analyses were performed using a two-stage linear mixed model and an additive genetic model. In Stage 1, residuals were obtained from a regression of fasting glucose on age, sex, and BMI. The resulting residuals were ranked and inverse normalized. In Stage 2, SNP association was tested by regressing the values from Stage 1 on imputed dosages, adjusted for three significant principal components obtained from the R package SNPRelate (version 1.28.0) [12] as fixed effects and cryptic relatedness as a random effect using the emmax test in EPACTS (version 3.3.0) [13]. The genome-wide significance level α was declared to be $5 \times 10^{-8}$. To test for secondary signals, the analysis in Stage 2 was repeated with the inclusion of genome-wide significant SNPs as covariates. R (version 4.0.3) was used in the analyses [14].

## Replication analysis

The Multi-Ethnic Study of Atherosclerosis (MESA) [15] and the Framingham Heart Study (FHS) [16] are prospective studies designed to identify risk factors for subclinical atherosclerosis. Individual-level genotype and phenotype data were obtained by authorized access to dbGaP. The China America Diabetes Mellitus (CADM) study is a case-control study of T2D in China [17]. The Africa America Diabetes Mellitus (AADM) study is a case-control study of type 2 diabetes in Africans [18, 19]. The Howard Family University Study (HUFS) is a population-based cross-sectional study of African Americans in Washington, D.C. [20].

For fasting glucose/type 2 diabetes analysis, we aggregated data from 2,204/2,314 European Americans, 632/697 Chinese Americans, 1,080/1290 Hispanic Americans, and 1,206/1,407 African Americans from MESA; 2,211/4,378 West Africans from AADM; 1,548/1,754 African Americans from HUFS; and 2,430/2,605 African Americans from ARIC; 985/1,883 Chinese from CADM; and 2,007/2,061 from FHS, totaling 14,303/18,389 individuals, respectively. Genotype data comprised approximately one million SNPs using the Affymetrix Genome-wide Human SNP Array 6.0 (ARIC, MESA, and HUFS) or two million SNPs using the Affymetrix Axiom Genome-wide PanAFR Array (AADM). Affymetrix Axiom® Exome Genotyping arrays (~ 300,000 markers) were used in CADM. The Illumina HumanOmni5M array (~4.3M markers) was used in FHS. For in-house data sets (AADM, HUFS, and CADM) for which we collected individual-level data from study participants, we performed sex checks. For the data sets from dbGaP (ARIC, FHS, and MESA), we relied on documentation available within dbGaP. Quality control, genotype imputation, transformation of fasting glucose levels, covariates (including three significant principal components and cryptic relatedness), and association testing were the same as for the discovery analysis. We performed inverse variance-weighted fixed effects meta-analysis using METAL [21]. Coordinates were based on the hg19 build.

## Variant annotation

We used SnpEff to annotate variant effects [22]. SnpEff integrates with other tools in sequencing data analysis pipelines and contains two steps, variant annotation and effect prediction. Variant annotation datasets were built using a reference genome (hg19). Two methods, SNAP [23] and I-Mutant3 [24], were used to assess discriminative power, a raw numerical score reflecting direction and reliability of the prediction, for each SNP. Discriminative power is the distance of the actual prediction to the decision boundary (score = 0), which reflects the reliability of the prediction and the severity of the predicted effects [25].

PolyPhen-2 is a tool that predicts the possible impact of an amino acid substitution on the structure and function of a human protein [26]. SIFT is a tool that predicts amino acid changes that affect protein function, distinguishing between functionally neutral and deleterious amino acid changes [27]. Combined Annotation Dependent Depletion (CADD) is a tool for scoring the deleteriousness of variants in the human genome [28, 29]. CADD integrates multiple annotations to generate scores that strongly correlate with allelic diversity, pathogenicity of both coding and non-coding variants, and experimentally measured regulatory effects and that highly rank causal variants. Polyphen-2, SIFT, and CADD scores were all retrieved from Ensembl 104 [30].

## Fine mapping

Region fine mapping was performed using the R package CAVIARBF (version 0.2.1), an approximate Bayesian method that can incorporate functional annotation [31]. Minimal data requirements are marginal statistical test results and linkage disequilibrium between SNPs. SNPs with MAF $\geq$ 0.05 within the gene region ±150kb were selected. SNP annotations were coded for the absence (0) or presence (1) of promoter histone marks, enhancer histone marks, DNAse I hypersensitive sites, or bound proteins as provided by HaploReg v4.1 [32]. Bayes factors were calculated conditional on a maximum number of causal SNPs. The estimated Bayes factors and prior probabilities were then used to estimate the posterior inclusion probabilities.

## Additive association evaluation

Linear regression and logistic regression were used to determine the joint additive effect across associated independent loci for fasting glucose levels and T2D status, respectively. Whereas rank-based transformations cannot be back transformed, we log-transformed fasting glucose levels in order to be able to obtain effect sizes in original units. We regressed traits on the number of effect alleles, with adjustment for age, BMI, sex, significant principal components (PCs) by study. The analysis was performed using SAS 9.4 (Cary, NC, USA). The R package meta (version 5.1) [33] was used for meta-analysis with an inverse variance-weighted fixed effects method.

## Trait loci annotation

An expression QTL (eQTL) is a genomic locus that affects expression levels of mRNA. A splicing QTL (sQTL) is a genomic locus that affects the expression of RNA isoforms generated by alternative splicing events. We retrieved data on eQTL and sQTL annotations from the Genotype-Tissue Expression (GTEx) Portal (https://gtexportal.org).

## Protein structure and function predictions

Based on the protein sequence-to-structure-to-function paradigm, we uploaded translated sequences to the I-TASSER online server (https://zhanggroup.org//I-TASSER/) [34–36].

I-TASSER uses template-based fragment assembly simulations of amino acid sequences to predict three-dimensional protein structures, which are then used to find matches in a protein function database to predict protein functions. The predicted protein structures were viewed and analyzed using PyMol [37].

### Ethics statement

Ethical approval for the AADM study was obtained from the National Institutes of Health, the Howard University Institutional Review Board, and from ethics committees in Ghana (University of Ghana Medical School Research Ethics Committee and Kwame Nkrumah University of Science and Technology Committee on Human Research Publication and Ethics), Kenya (Moi Teaching & Referral Hospital/Moi University College of Health Sciences Institutional Research and Ethics Committee), and Nigeria (National Health Research Ethics Committee of Nigeria). Ethical approval for HUFS was obtained from the Howard University Institutional Review Board. Ethical approval for CADM was obtained from the institutional review boards of Howard University, National Institutes of Health, and Suizhou Central Hospital (Suizhou, China). Written informed consent was obtained from each participant. All clinical investigation was conducted according to the principles expressed on the Declaration of Helsinki.

## Results

Phenotyping, genotyping, and imputation summaries for all discovery and replication studies are presented in S1 Table and S1 and S2 Figs. Within individuals of European ancestry, males had higher BMI than females. In contrast, BMI was higher in females than males among Africans, African Americans, and Hispanic Americans. Males had higher fasting glucose levels than females in all groups.

The discovery and replication analyses included totals of 9,232 and 14,303 individuals, respectively. Three loci reached genome-wide significance (Fig 1 and S2 Table). The genomic control variance inflation factor indicated no inflation due to population stratification (l = 1.01; S3 Fig). Two of the lead SNPs were missense mutations and the third lead SNP was intronic (Table 1). Regional association and Bayesian fine mapping indicated that rs1260326 (*GCKR*), rs560887 (*G6PC2*), and rs13266634 (*SLC30A8*) had the highest marginal posterior inclusion probabilities (PIP) in their respective loci (S4 Fig). Conditional on the lead SNPs rs1260326 (*GCKR*), rs560887 (*G6PC2*), and rs13266634 (*SLC30A8*), no signal remained (S5 Fig). The effect alleles at all three lead SNPs were associated with lower fasting glucose (Table 1). The associations at all three lead SNPs were replicated in overall meta-analysis (S3 Table). Of the two suggestive loci (Fig 1, $5 \times 10^{-7} \leq p < 5 \times 10^{-8}$), only the association at the locus on chromosome 11 was replicated (S4 Table).

The variant rs1260326 (*GCKR*) is a missense mutation, resulting in a substitution from leucine to proline at position 446. The coding effect of rs1260326 was estimated by SnpEff as moderately important (Table 1) and annotated as tolerated by SIFT and benign by Polyphen-2 [30]. Position 446 in GCKR is located at the interface with GCK (Fig 2). L446 is closer to the middle of the interface whereas L446P is closer to GCK (S6 Fig). The variant rs560887 (*G6PC2*) is intronic and estimated to have low impact (Table 1). The variant rs13266634 (*SLC30A8*) is a missense mutation, resulting in a substitution from arginine to tryptophan at position 325, annotated as a moderate change by SnpEff (Table 1) and as tolerated by SIFT and benign by PolyPhen-2. I-TASSER predicted four possible protein structures based on an amino sequence with R325W. The four predicted protein structures were similar to each other, but all were different from wild type (Fig 3) and consistent with a moderate change in protein structure.

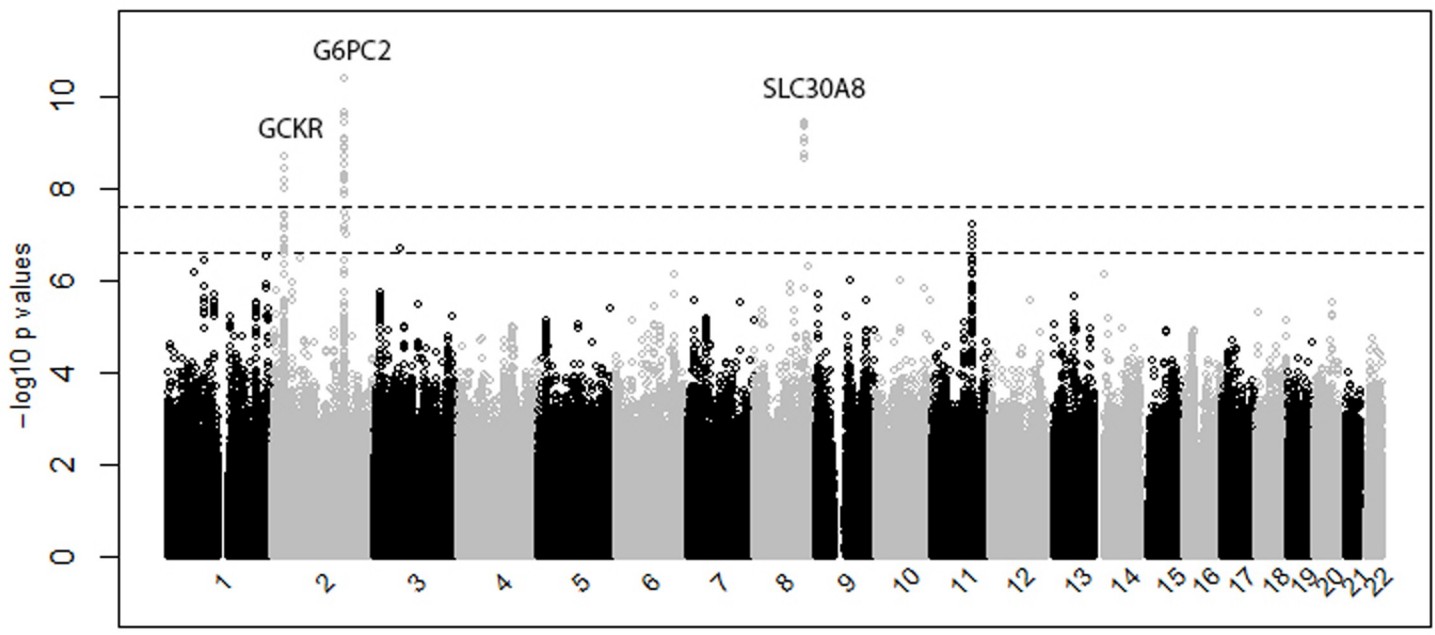

**Fig 1. Manhattan plot for discovery analysis based on individuals with European ancestry.** The *x*-axis represents chromosomal positions, and the *y*-axis represents -$\log_{10}$(*p*-value). The two dotted lines represent -$\log_{10}$ ($5\times10^{-8}$) and -$\log_{10}$ ($5\times10^{-7}$), respectively.

To determine a best fit model jointly across loci, the three loci and all possible interactions were specified in a full model. Regression with backward selection (SLSTAY = 0.10) was used to eliminate variables (S5 Table). The final model included the three lead SNPs without any interactions. After excluding possible interactions, we found that the effect alleles influence fasting glucose in an additive manner (Fig 4). For each copy of a *T* allele at any of the three SNPs, an additive effect of -0.012 mmol/L on fasting glucose was identified in the discovery sample ($p = 3.0\times10^{-28}$) and replicated in trans-ethnic meta-analysis ($n = 14,303$, $\beta = -0.0088$, SE = 0.0011, $p = 2.15\times10^{-16}$, S7 Fig). We also estimated the joint additive effect of the three SNPs on the risk of T2D in a total of 28,120 individuals with ($n = 4,585$) or without ($n = 23,535$) T2D. The three SNPs were associated with significantly reduced T2D risk, with an odds ratio of 0.93 (95% confidence interval [0.88, 0.98], $p = 0.0062$, S8 Fig). Notably, none of the individuals with 6 *T* alleles had T2D, compared to 27% of those with 0 *T* alleles (Fig 5).

**Table 1. Lead SNPs from discovery GWAS.**

| Chromo-some | Position (bp) | Gene | SNP | Reference/ Alternate Allele | Annotation | Alternate Allele Frequency | Beta | Standard Error | P-value | Variance explained | Impact | Sequence Change |
|---|---|---|---|---|---|---|---|---|---|---|---|---|
| 2 | 27730940 | *GCKR* | rs1260326 | C/T | Nonsynonymous | 0.4082 | -0.0113 | 0.0020 | 1.06E-08 | 0.0035 | MODERATE | c.1337T>C, p.Leu446Pro |
| 2 | 169763148 | *G6PC2* | rs560887 | C/T | Intron | 0.3026 | -0.0132 | 0.0020 | 4.39E-11 | 0.0047 | LOW | c.441-26T>C |
| 8 | 118184783 | *SLC30A8* | rs13266634 | C/T | Nonsynonymous | 0.3159 | -0.0120 | 0.0019 | 4.28E-10 | 0.0042 | MODERATE | c.973C>T, p.Arg325Trp |

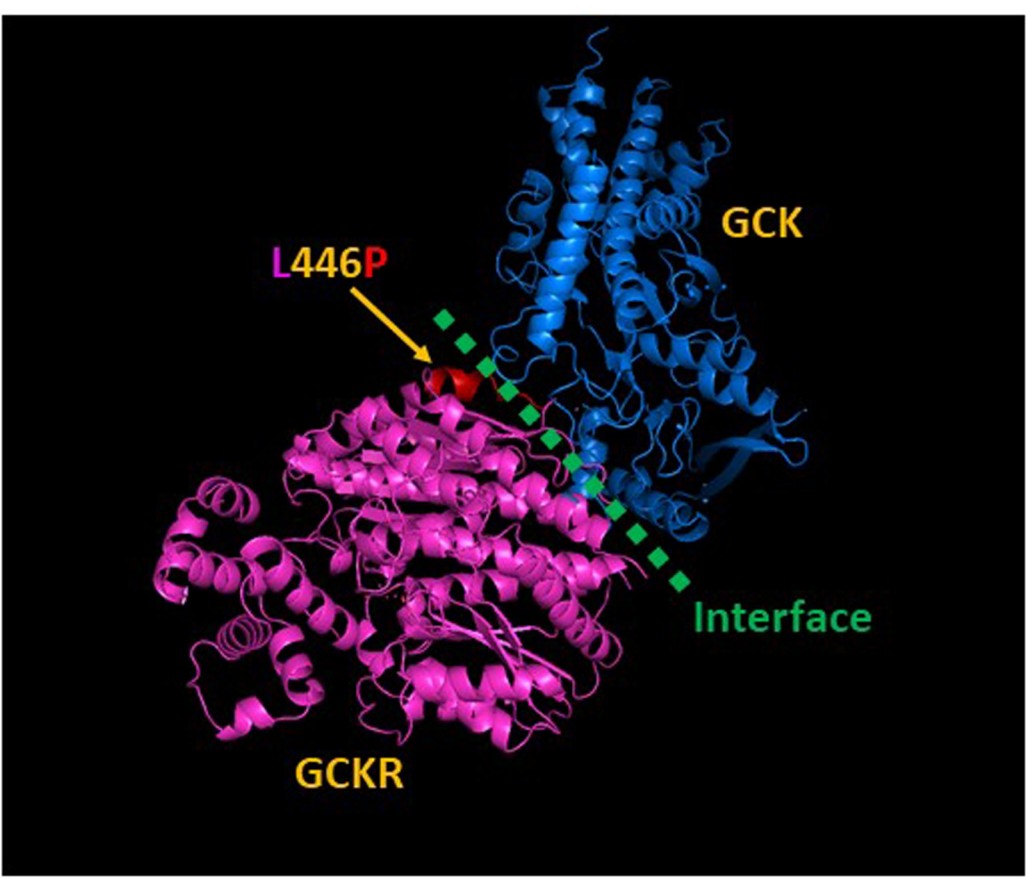

**Fig 2. Wild type GCKR protein (pink) interacts with wild type GCK protein (blue).** The position of interaction in
GCKR is L446 (rs1260326, red). Green dotted line presents the proximity of the interface between GCK and GCKR.

## Discussion

Based on a genome-wide analysis of fasting glucose, we identified three loci (*GCKR*, *G6PC2*,
and *SLC30A8)* that are involved in glucose regulation as previously reported [4, 38]. Here, we
showed that the joint effect of these loci was associated with lower fasting glucose levels as well
as lower risk of T2D. The missense SNP rs1260326 in *GCKR* is significantly associated with
fasting glucose in non-diabetic and untreated diabetic individuals with European ancestry.
This association was replicated in trans-ethnic meta-analysis of European Americans, Chinese,
Chinese Americans, Hispanic Americans, African Americans, and Africans. The SNP
rs1260326 has been associated with fatty liver, triglycerides, and very low-density lipoprotein
cholesterol in obese children and adolescents [39]. The position in GCKR changed by
rs1260326 interacts with GCK; the mutation leads to reduced capability to response to fruc-
tose-6-phosphate, increased *GCK* activity in the liver, and reduced glucose levels [40–42].

*G6PC2* (rs560887) has been reported to be associated with fasting glucose [40, 43] and with
the 30 min. incremental insulin response in the oral glucose tolerance test [44]. The encoded
protein allows the release of glucose into the bloodstream. rs560887 is an expression QTL
(eQTL) for *G6PC2* in several tissues but most strongly in subcutaneous adipose tissue, with the
alternate allele associated with lower gene expression (S6 Table). It is also a splicing QTL
(sQTL) for *NOSTRIN* in several tissues (S7 Table). NOSTRIN binds the enzyme responsible
for production of nitric oxide, which is involved in neurotransmission, inflammatory

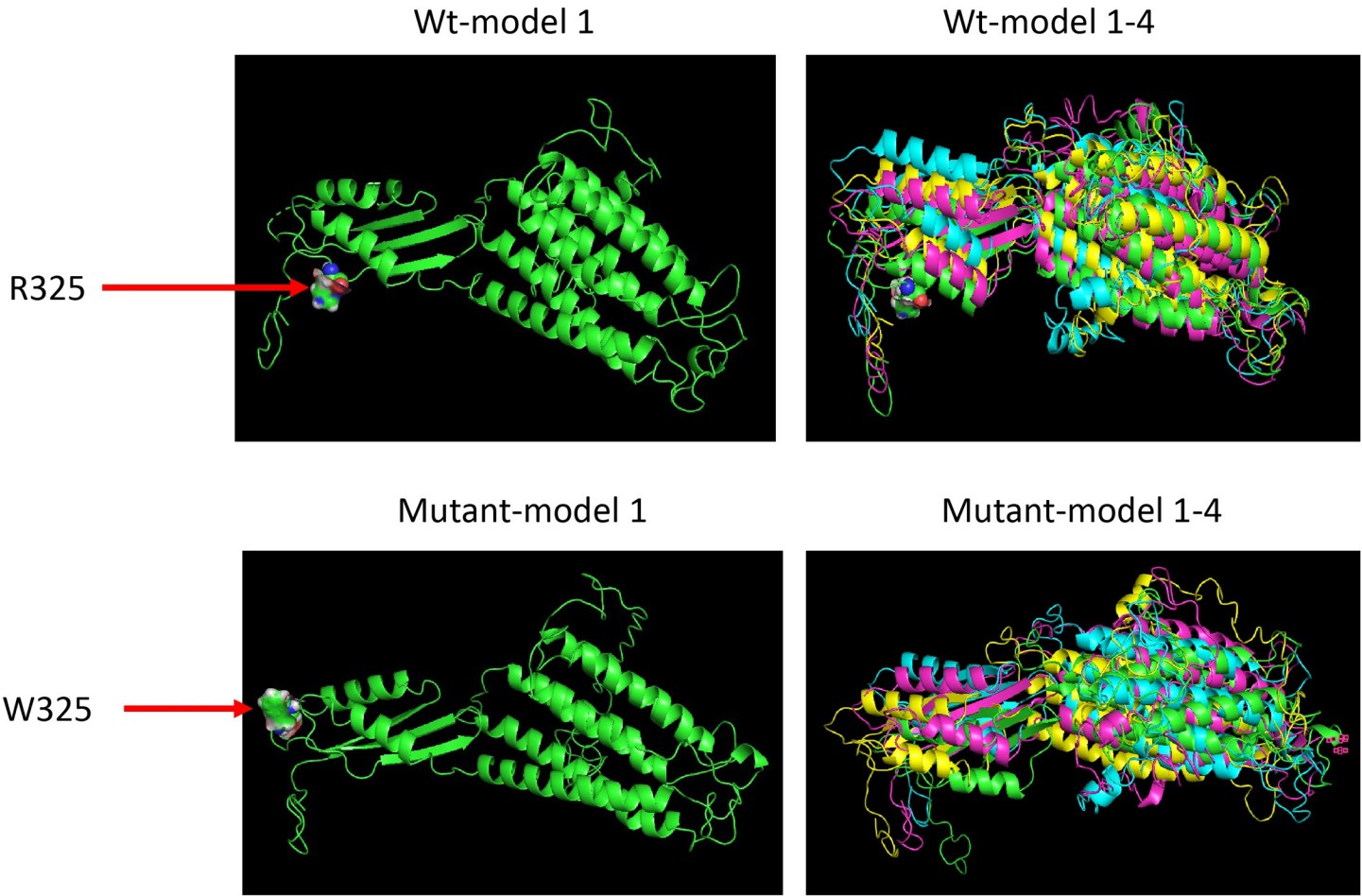

**Fig 3.** The SLC30A8 protein structures for Wild Type (Wt, top) and R325W (rs1326634, bottom). Each amino acid sequence yielded four predicted protein structures called models 1 to 4 for Wt and mutant, respectively. Wt-Model 1 (top left) is the 1st 3D structure predicted by comparative molecular modeling through I-TASSER. Wt-Model 1–4 shows the overlap of the four predicted 3D structures for SLC30A8 wild type (top right). The mutant structures (bottom) are labeled correspondingly.

responses, and vascular homeostasis [45]. An effect on *NOSTRIN* could explain the association of rs560887 with pulse pressure and other phenotypes [46]. The SNP rs560887 is in strong LD with rs573225 ($r^2 = 0.90$) in EUR but weaker LD in AFR ($r^2 = 0.60$) [32]. rs573225 is 207 bp upstream of *G6PC2*. Like rs560887, rs573225 was associated with lower fasting glucose (β = -0.010, SE = 0.002, $p = 6.57 \times 10^{-8}$) in our discovery study and was replicated (β = -0.005, SE = 0.0018, $p = 0.0031$). Also like rs560887, rs573225 is an eQTL for *G6PC2* (S6 Table) and an sQTL for *NOSTRIN* (S7 Table). However, rs573225 has a phred-scaled CADD score of 15.97, compared to 0.210 for rs560887, indicating that rs573225 is more strongly deleterious than rs560887 [30]. rs573225 maps to the highly conserved 2nd position of a predicted regulatory motif for HNF4, with the alternate allele associated with weaker binding of HNF4 [32] and lower expression of *G6PC2* (S6 Table). Thus, annotations not included in the fine mapping analysis (specifically, CADD scores and predicted regulatory motifs) provide evidence that rs573225 might be a better candidate causal variant and that rs560887 might simply be tagging rs573225.

We found that the missense variant rs13266634 in *SLC30A8* was associated with fasting glucose levels and was previously reported to be associated with T2D risk as well as glucose and proinsulin levels [3, 47]. The *T* allele at rs13266634 is associated with enhanced insulin

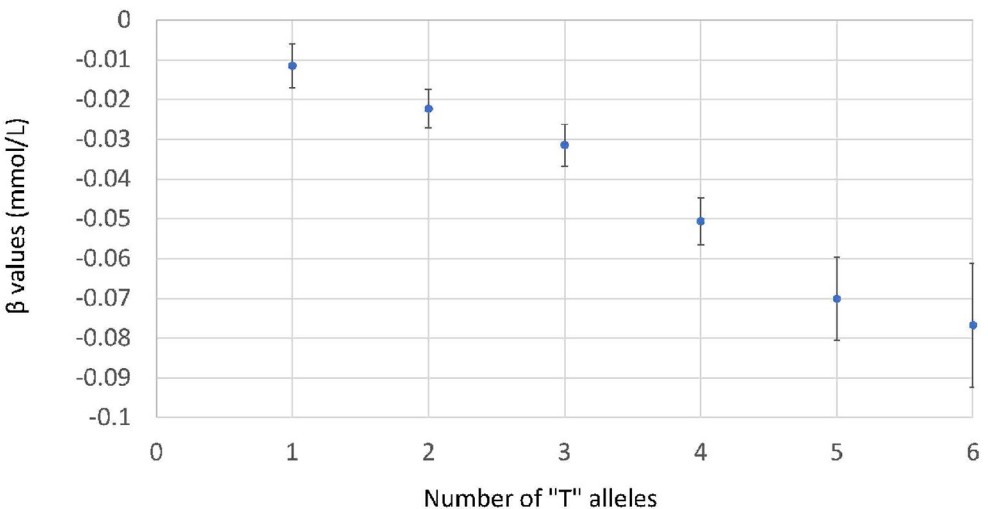

**Fig 4. Joint effect size and standard error for fasting glucose at rs1260326 (*GCKR*), rs560887 (*G6PC2*), and rs13266634 (*SLC30A8*).** The reference group is homozygous for the reference allele at all three SNPs. At each SNP, the *T* allele is the allele associated with lower fasting glucose.

secretion from pancreatic β cells and inhibited hepatic insulin clearance, leading to increased peripheral insulin levels and decreased peripheral glucose levels [48]. SLC30A8 is a transmembrane transporter, with the ligand zinc binding to a histidine-rich region from positions 197 to 205 [49]. The position in SLC30A8 changed by rs13255534, position 325, is located on the surface of the protein and maps to the cytoplasmic tail at a point where the protein bends back on itself [49]. Therefore, rs13266634 might not affect binding affinity but might affect either protein stability or interaction with other cytoplasmic components of the transport process.

Functional studies that follow up on findings of genetic associations are critical. One way to assess function is based on analysis of predicted amino acid sequences [50]. Two of the three genetic variants identified in our study were missense. Wild type and mutant amino acid

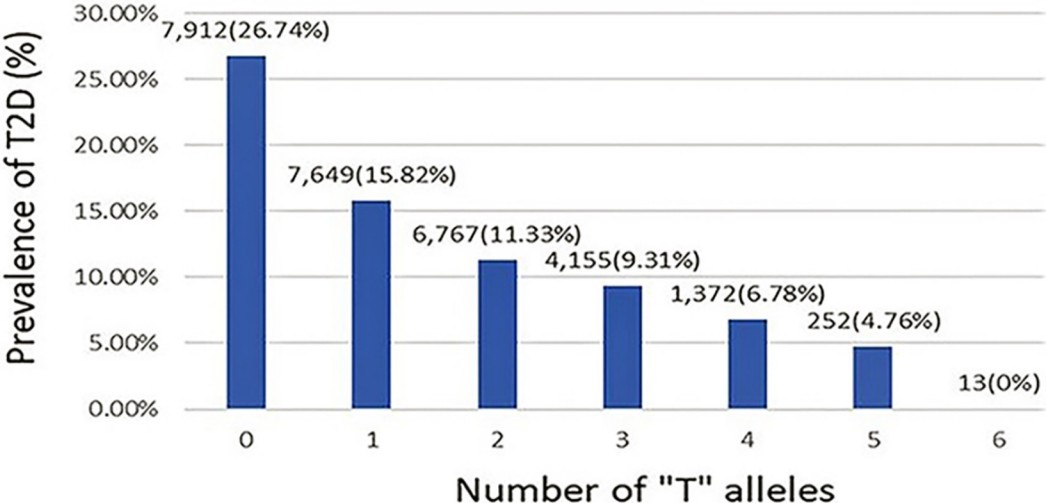

**Fig 5. Joint effect size for the prevalence of T2D at rs1260326 (*GCKR*), rs560887 (*G6PC2*), and rs13266634 (*SLC30A8*).** The label above each bar provides the number of individuals (% prevalence of T2D). At each SNP, the *T* allele is the allele associated with protection against T2D.

sequences were uploaded onto the I-TASSER server and predicted protein structures were imported into PyMOL for predicted protein function. Moderate protein structure differences were predicted at both rs1260326 (*GCKR*) and rs13266634 (*SLC30A8*), leading to predicted changes in protein function. The protein structures modeled by I-TASSER suggest that both rs1260326 and rs13266634 have the potential to change the corresponding protein structures and functions, which might result in altered glucose levels. The predicted structure of GCKR revealed that position 446 is located at the proximity of the interface between GCK and GCKR; therefore, L446P could affect the relative positioning of GCK and GCRK at the interface. This alteration could potentially impact the interaction efficiency of the two proteins, which can be assessed *in vitro* through either immunoprecipitation or fluorescence resonance energy transfer. Mutations in mice can be created using CRISPR editing technology so that the functional impacts of both GCKR-L446P and SLC30A8-R325W mutations could be tested *in vivo*. Structural information can also facilitate the rational design and development of targeted drugs and antibodies.

An intergenic locus on chromosome 11 33.4 kb upstream of *MTNR1B* reached suggestive levels of significance in the discovery study and was replicated. There are two variants with $r^2 \geq 0.8$ in Europeans for the lead SNP rs6483204: rs3847554 and rs6483205 [32]. The variant rs3847554 has been previously reported as associated with fasting plasma glucose [51], but the association at rs3847554 did not replicate in our study due to heterogeneous effect sizes. The variants rs6843204 and rs3847554 are eQTLs for *SLC36A4* in esophagus mucosa. SLC36A4 is a non-proton-coupled amino acid transporter. There is no evidence based on histone marks, proteins bound, or binding motifs that rs6483204 could be causal [32]. For rs3847554, the only evidence is a change in a binding motif for CDCL5 [32].

## Conclusions

We analyzed GWAS data with 23,535 individuals, either nondiabetics or untreated diabetics, and identified and replicated three independent SNPs in *GCKR*, *G6PC2*, and *SLC30A8* associated with fasting glucose levels. Each copy of the alternate allele at any of these three SNPs was associated with a reduction of 0.012 mmol/L in fasting glucose. The alternate allele at rs1260326 (*GCKR*) is associated with increased glycolysis, the alternate allele at rs560887 (*G6PC2*) is associated with decreased gluconeogenesis, and the alternate allele at rs13266634 (*SLC30A8*) is associated with increased insulin secretion. Each copy of the alternate allele at any of the three SNPs was associated with a 7% reduced risk of T2D, indicating that the associations are not just physiologically relevant but also pathophysiologically relevant.

## Supporting information

**S1 Fig. Density plots for fasting glucose (mmol/L) in the discovery study.** Untransformed (left) and log-transformed (right).
(PDF)

**S2 Fig. Density plots for log transformed fasting glucose (mmol/L) in the replication studies.** WHT, CHI, HIS, and AA refer to European Americans, Chinese, Hispanic Americans, and African Americans, respectively.
(PDF)

**S3 Fig. Quantile-quantile plot of *p*-values for fasting glucose levels in individuals with European ancestry in ARIC.** The *x*-axis represents expected *p*-values, and the *y*-axis represents observed *p*-values. All *p*-values are transformed as $-\log_{10}(p\text{-value})$.
(PDF)

**S4 Fig. Three panels represent *GCKR*, *G6PC2*, and *SLC30A8*, respectively.** (Top) Region association plot: The *x*-axis represents position in Mb. The *y*-axis represents -$\log_{10}$ *p*-values. Sky-blue lines represent recombination rates (cM/Mb) from the 1000 Genomes Project. (Bottom) Posterior inclusion probabilities (PIP) based on fine mapping. The *x*-axis represents position in Mb. The *y*-axis represents PIP values.
(PDF)

**S5 Fig. Genome-wide conditional analysis of fasting glucose in individuals with European ancestry.** Row 1: Conditioning on rs1260326 (*GCKR*) abolished the peak at *GCKR*. Row2: Conditioning on rs1260326 (*GCKR*) and rs13266634 (*SLC30A8*) abolished the peaks at *GCKR* and *SLC30A8*. Row 3: Conditioning on rs1260326 (*GCKR*), rs560887 (*G6PC2*), and rs13266634 (*SLC30A8*) eliminated all genome-wide significant signals.
(PDF)

**S6 Fig. Model structure of GCK with GCKR wild type (green) and L446P mutant (red) complexes.**
(PDF)

**S7 Fig. Forest plot from meta-analysis of fasting glucose levels in nine replication studies (*n* = 14,303).**
(PDF)

**S8 Fig. Forest plot from meta-analysis of risk of T2D in discovery and replication studies (*n* = 28,120).**
(PDF)

**S1 Table. Study characteristics for discovery and replication studies.**
(XLSX)

**S2 Table. Genome-wide significant association results in the discovery analysis.**
(XLSX)

**S3 Table. Trans-ethnic replication analysis results.**
(XLSX)

**S4 Table. Genome-wide suggestive association results.**
(XLSX)

**S5 Table. Parameter estimation in backward regression.**
(XLSX)

**S6 Table. GTEx expression QTL data.**
(XLSX)

**S7 Table. GTEx splicing QTL data.**
(XLSX)

## Acknowledgments

The Atherosclerosis Risk in Communities study has been funded in whole or in part with federal funds from the National Heart, Lung, and Blood Institute, National Institute of Health, Department of Health and Human Services, under contract numbers HHSN268201700001I, HHSN268201700002I, HHSN268201700003I, HHSN268201700004I, and HHSN268201700005I. The authors thank the staff and participants of the ARIC study for their important contributions. Funding for ARIC Gene Environment Association Studies

(GENEVA) was provided by National Human Genome Research Institute grant U01HG004402 (E. Boerwinkle). The datasets used for the analyses in this manuscript were obtained from dbGaP through dbGaP accession study number phs000280.v5.p1. MESA and the MESA SHARe project are conducted and supported by the National Heart, Lung, and Blood Institute (NHLBI) in collaboration with MESA investigators. Support for MESA is provided by contracts N01-HC95159, N01-HC-95160, N01-HC-95161, N01-HC-95162, N01-HC-95163, N01-HC-95164, N01-HC-95165, N01-HC95166, N01-HC-95167, N01-HC-95168, N01-HC-95169, UL1-RR-025005, and UL1-TR-000040. Funding for SHARe genotyping was provided by NHLBI Contract N02-HL-64278. Genotyping was performed at Affymetrix (Santa Clara, California, USA) and the Broad Institute of Harvard and MIT (Boston, Massachusetts, USA) using the Affymetrix Genome-Wide Human SNP Array 6.0. This manuscript was not prepared in collaboration with MESA investigators and does not necessarily reflect the opinions or views of MESA, or the NHLBI. The datasets used for the analyses in this manuscript were obtained from dbGaP through dbGaP accession study number phs000209.v13.p3. The Framingham Heart Study is conducted and supported by the National Heart, Lung, and Blood Institute (NHLBI) in collaboration with Boston University (Contract No. N01-HC-25195, HHSN268201500001I and 75N92019D00031). This manuscript was not prepared in collaboration with investigators of the Framingham Heart Study and does not necessarily reflect the opinions or views of the Framingham Heart Study, Boston University, or NHLBI. The datasets used for the analyses in this manuscript were obtained from dbGaP through dbGaP accession study number phs000007.v32.p13. The Genotype-Tissue Expression (GTEx) Project was supported by the Common Fund of the Office of the Director of the National Institutes of Health, and by NCI, NHGRI, NHLBI, NIDA, NIMH, and NINDS. The data used for the analyses described in this manuscript were obtained from the GTEx Portal (V8) on 04/10/21. The Howard University Family Study (HUFS), the China America Diabetes Mellitus (CADM) study, and the Africa America Diabetes Mellitus (AADM) study are in-house studies and supported by the National Human Genome Research Institute, National Institutes of Health. This work utilized the computational resources of the NIH HPC Biowulf cluster (https://hpc.nih.gov). The contents of this publication are solely the responsibility of the authors and do not necessarily represent the official view of the National Institutes of Health. This research was supported by the Intramural Research Program of the Center for Research on Genomics and Global Health (CRGGH). The CRGGH is supported by the National Human Genome Research Institute, the National Institute of Diabetes and Digestive and Kidney Diseases, the Center for Information Technology, and the Office of the Director at the National Institutes of Health (1ZIAHG200362).

## Author Contributions

**Conceptualization:** Guanjie Chen.

**Data curation:** Guanjie Chen.

**Formal analysis:** Guanjie Chen.

**Investigation:** Guanjie Chen, Ayo P. Doumatey, Charles N. Rotimi.

**Methodology:** Guanjie Chen, Daniel Shriner, Jie Zhou.

**Project administration:** Guanjie Chen.

**Resources:** Guanjie Chen, Daniel Shriner, Jie Zhou, Poorni Adikaram, Amy R. Bentley.

**Software:** Guanjie Chen, Daniel Shriner, Jianhua Zhang, Jie Zhou, Poorni Adikaram, Adebowale Adeyemo.

**Supervision:** Adebowale Adeyemo, Charles N. Rotimi.

**Validation:** Guanjie Chen, Jie Zhou, Ayo P. Doumatey, Adebowale Adeyemo.

**Visualization:** Guanjie Chen, Jianhua Zhang, Jie Zhou, Ayo P. Doumatey, Adebowale Adeyemo.

**Writing – original draft:** Guanjie Chen, Jianhua Zhang, Amy R. Bentley, Charles N. Rotimi.

**Writing – review & editing:** Guanjie Chen, Daniel Shriner, Jianhua Zhang, Amy R. Bentley, Adebowale Adeyemo, Charles N. Rotimi.

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
