## [Decision Letter · Decision Letter 0]

25 Nov 2021

PONE-D-21-24235Additive genetic effect of GCKR, G6PC2, and SLC30A8 variants on fasting glucose levels and risk of type 2 diabetesPLOS ONE

Dear Dr. Chen,

Thank you for submitting your manuscript to PLOS ONE. After careful consideration, we feel that it has merit but does not fully meet PLOS ONE’s publication criteria as it currently stands. Therefore, we invite you to submit a revised version of the manuscript that addresses the points raised during the review process.

Specifically, the Reviewers have raised a number of concerns related to the methods and results that require further clarification in the text. 

We look forward to receiving your revised manuscript.

Kind regards,

Nicholette D. Palmer, Ph.D.

Academic Editor

PLOS ONE

Journal Requirements:

2. Thank you for including your ethics statement:  "All in-house studies received ethics approval from Institutional Review Boards and written consent was obtained from each participant.".  

a. Please amend your current ethics statement to include 1) the full name of the ethics committee/institutional review board(s) that approved the specific study/studies for which data collection was conducted by the authors 2) details as to whether consent was informed

“The Atherosclerosis Risk in Communities study has been funded in whole or in part with federal funds from the National Heart, Lung, and Blood Institute, National Institute of Health, Department of Health and Human Services, under contract numbers HHSN268201700001I, HHSN268201700002I, HHSN268201700003I, HHSN268201700004I, and HHSN268201700005I.”

“This research was supported by the Intramural Research Program of the Center for Research on Genomics and Global Health (CRGGH). The CRGGH is supported by the National Human Genome Research Institute, the National Institute of Diabetes and Digestive and Kidney Diseases, the Center for Information Technology, and the Office of the Director at the National Institutes of Health (1ZIAHG200362).”

Reviewers' comments:

Reviewer's Responses to Questions

**Comments to the Author**

1. Is the manuscript technically sound, and do the data support the conclusions?

Reviewer #1: Yes

Reviewer #2: Yes

2. Has the statistical analysis been performed appropriately and rigorously? 

Reviewer #1: Yes

Reviewer #2: Yes

3. Have the authors made all data underlying the findings in their manuscript fully available?

Reviewer #1: Yes

Reviewer #2: No

4. Is the manuscript presented in an intelligible fashion and written in standard English?

Reviewer #1: Yes

Reviewer #2: Yes

5. Review Comments to the Author

Reviewer #1: Chen et al conducted GWAS with 23,535 individuals, either nondiabetics or

298 untreated diabetics, and identified and replicated three independent SNPs in GCKR, G6PC2, and SLC30A8 associated with fasting glucose levels.

Here are my comments:

1. In Abstract line24, authors should clarify whether the they have multiple outcomes in the regression model. If it is only one outcome in the model, the 'multivariate' should be 'multivariable'.

2. In the association analysis part, The fasting glucose is log-transformed. It seems the density plot still have tail after transformation. I suggest authors check the normal test such as Shapiro–Wilk test before and after transformation.

3. If the log-transformation for FG works, the residual should follow normal distribution. If this is the case, author should provide the reason why they also need the inverse normal transformation for the residuals.

4. How was the distribution of FG in the replication?

5. Authors should describe the statistical method for T2D.

6. In line 203, author should include the backward selection criteria.

Reviewer #2: The Chen et al performed a GWAS of fasting glucose in non-diabetic and untreated diabetic individuals of European ancestry. 3 loci displayed significant associations (SNPs with P<5x10-8). The association of the 3 lead SNPs (after fine mapping) was replicated in trans-ethnic meta-analysis of European, Chinese, Hispanic and African ancestry. The authors further test the possible consequences of non-synonymous SNPs using various in silico tools. Finally, a genetic score combining the FPG reducing alleles was derived. The score was associated with FPG and T2D.

The manuscript is interesting and easy to read. I have the following comments/suggestions:

Methods :

- in replications studies that are family studies (eg, FHS and HUFS), did you only keep unrelated individuals as in discovery ?

- I suppose all the alleles shown in the manuscript tables/text are on the + strand ? please specify.

- Replication studies : please clearly state the genome build used in methods and tables (same as discovery study so, hg19 ? ).

- All R packages: provide version.

- Reference the R software and provide version.

Row 82: Is there a missing "." here ? if not, the sentence is not really clear.

Row 83: "Among Europeans" : please specify how the EUR ancestry was determined (self report, genetic data ? or both ?).

Row 88: use "race and ethnicity" instead or "race" only.

Row 89: T2D definitions in row 79 and 91 are slightly different, please clarify.

Row 96: GWAS QC: no sex check control performed? no ethnicity check? Also maybe provide some references for the QC of replication studies?

Row 107: Can the author provide a brief justification as to why they used a two-stage mixed model? (i.e. first regressing on Age sex BMI on log FPG, then use ranked inverse normalized residuals for stage 2? Also, does this mean that all FPG related results (Beta, SE, etc) we see in the tables are back transformed and are in mmol/l? if yes, this should be mentioned.

Row 111: How many PCs adjusted for?

Row 118: What was the random factor in the mixed models?

Row 118-119: Please provide a full dbGAP accession number with the version (phsxxxxxx.vx.px format).

Row 121: Add citation to the dbGAP database itself.

Row 126: Consider giving the number of case controls by ethnicity form each study in a supplementary table.

Row 131: For each study (both stage 1 and replication studies) and for each ethnic group, provide a supplementary table with genotyping array, imputation panel used, filtering criteria (e.g. info score and MAF) and the resulting number of SNPs used (in the analysis in addition to the paragraph lines 131 to 138).

Row 164: Add ref to R package Meta.

Row 152: How do you define the presence of "LD between SNP".

Results:

Row 188: "The associations at all three lead SNPs were replicated", please specify that this is for the overall Meta-A, and briefly describe the replication results by ethnic groups.

Row 188: "Of the two suggestive loci": Please specify what you mean by suggestive locus ( 5*10-7< P <5*10-8 ?)

Row 192 + 198: SIFT, Polyphen-2 and CADD not mentioned at all in the methods. Please correct. also Ensembl is referenced instead of the programs and original papers themselves, if SIFT, Polyphen-2 and CADD results were extracted from Ensembl, then this should be specified in the methods, and references should include the original SIFT, Polyphen-2 and CADD papers in addition to Ensembl.

row 193 + Figure S5: "The model protein structure of GCKR with L446P predicted an altered interaction between GCKR and GCK Figure S5)": not very clear from the figure S5 how the "interaction is altered", do you have a metric you use? a prediction P-value? also, maybe you could add arrows that point to where we're supposed to look on Figure S5 + specify what the purple color means?

Row 230: eQTL and sQTL analysis not described in methods and results.

Row 209: Please correct "nomarl".

Row 209: "compared with normal values are less ...". Please consider changing the phrasing.

Row 210: Please correct "mmol/".

Discussion:

Row 236: it is a bit confusing that authors describe rs560887 as lead SNP after fine mapping, having a low functional impact (row 195), but then discuss eQTL and sQTL effects in the discussion, and later suggest that rs573225 might be a better candidate. Was rs573225 included in fine mapping? what was its posterior probability? has the fine mapping performed in the different ethnic groups or restricted to EUR?

Row 258-261: " based on the sequence to structure... for predicted protein function": should not be included in discussion, it reads more like a methods paragraph.

Introduction:

Row 37: Please replace ".." by "."

Abstract:

Row 34: The authors should not just mention the results of 0 vs 6 T alleles and emphasize the gradual change of outcome based on the number of risk alleles (linear trend) instead.

Row 37: "Since non of the individuals homozygous for the ... identify individuals at low vs. high risk of developing type 2 diabetes": Not sure about this statement... I think if you derive a genetic score using more than 3 SNPs it will have a far better predictive power than a score with 3 SNPs.

All Tables, Figures and Supplementary Figures: Please add abbreviations and units when applicable.

Table S3:

-Add an Ethnicity column.

-Show meta-analysis heterogeneity results.

-Show meta-A results by ethnic group (Europeans, Africans + African descent, Chinese, Latinos)

Figure S3: Figure Resolution too low.

Figure S6: The figure title does not reflect the fact that you are testing the number of protective alleles.

Figure S7: The figure title does not reflect the fact that you are re testing the number of protective alleles.

References: I believe I found a couple of references that do not match what is being said in the main text, please double check the references.

6. PLOS authors have the option to publish the peer review history of their article (what does this mean?). If published, this will include your full peer review and any attached files.

Reviewer #1: No

Reviewer #2: **Yes: **Amel Lamri

---

## [Author Response · Author response to Decision Letter 0]

26 Jan 2022

Response to editor

Response: We revised the manuscript following the PLOS ONE style templates.

2. Thank you for including your ethics statement: "All in-house studies received ethics approval from Institutional Review Boards and written consent was obtained from each participant.".

a. Please amend your current ethics statement to include 1) the full name of the ethics committee/institutional review board(s) that approved the specific study/studies for which data collection was conducted by the authors 2) details as to whether consent was informed.

Response: The ethics statement has been amended as requested (lines 186-198).

Response: The amended ethics statement now appears in the Ethics Statement field of the submission form.

“The Atherosclerosis Risk in Communities study has been funded in whole or in part with federal funds from the National Heart, Lung, and Blood Institute, National Institute of Health, Department of Health and Human Services, under contract numbers HHSN268201700001I, HHSN268201700002I, HHSN268201700003I, HHSN268201700004I, and HHSN268201700005I.”

“This research was supported by the Intramural Research Program of the Center for Research on Genomics and Global Health (CRGGH). The CRGGH is supported by the National Human Genome Research Institute, the National Institute of Diabetes and Digestive and Kidney Diseases, the Center for Information Technology, and the Office of the Director at the National Institutes of Health (grant 1ZIAHG200362 to C.N.R.). The funders had no role in study design, data collection and analysis, decision to publish, or preparation of the manuscript.”

Response: All funding statements in the Acknowledgements section are acknowledgements required by dbGaP. We did nothing to obtain any of those grants nor did any of those grants fund our study. The current Funding Statement is correct, and the funding information provided in the current Funding Statement is the only information that should be published.

Response: The ethics statement appears only in the Materials and methods section (lines 186-198).

Response to reviewers

Reviewer #1: Chen et al conducted GWAS with 23,535 individuals, either nondiabetics or

298 untreated diabetics, and identified and replicated three independent SNPs in GCKR, G6PC2, and SLC30A8 associated with fasting glucose levels.

Here are my comments:

1. In Abstract line24, authors should clarify whether the they have multiple outcomes in the regression model. If it is only one outcome in the model, the 'multivariate' should be 'multivariable'.

Response: As there is only one outcome, we have changed multivariate to multivariable (line 22).

2. In the association analysis part, The fasting glucose is log-transformed. It seems the density plot still have tail after transformation. I suggest authors check the normal test such as Shapiro–Wilk test before and after transformation.

Response: The GWAS analysis utilized the rank-based inverse normalized transformation (line 102). This transformation does not depend on log transformation. The distribution of fasting glucose values is right-skewed because we included untreated T2D cases. Shapiro-Wilk normality testing is limited to a sample size of 5000 in R and 2000 in SAS. The Kolmogorov-Smirnov test is also used for normality testing, but we did not present results of that test because it has low power (PMID 23843808).

3. If the log-transformation for FG works, the residual should follow normal distribution. If this is the case, author should provide the reason why they also need the inverse normal transformation for the residuals.

Response: We used the rank-based inverse normal transformation of fasting glucose values in the GWAS analysis (line 102). We note that transformation of the dependent variable does not guarantee that residuals are normally distributed.

4. How was the distribution of FG in the replication?

Response: We added density plots for log-transformed fasting glucose for all replication studies in S2 Fig (line 201).

5. Authors should describe the statistical method for T2D.

Response: We detailed the statistical association method (logistic regression) for T2D (lines 163-164).

6. In line 203, author should include the backward selection criteria.

Response: We added that the select stay = 0.10 (SLSTAY = 0.10) criterion was used for backward selection (lines 247-248).

Reviewer #2: The Chen et al performed a GWAS of fasting glucose in non-diabetic and untreated diabetic individuals of European ancestry. 3 loci displayed significant associations (SNPs with P<5x10-8). The association of the 3 lead SNPs (after fine mapping) was replicated in trans-ethnic meta-analysis of European, Chinese, Hispanic and African ancestry. The authors further test the possible consequences of non-synonymous SNPs using various in silico tools. Finally, a genetic score combining the FPG reducing alleles was derived. The score was associated with FPG and T2D.

The manuscript is interesting and easy to read. I have the following comments/suggestions:

Methods:

- in replications studies that are family studies (eg, FHS and HUFS), did you only keep unrelated individuals as in discovery?

Response: In the discovery analysis, we did not keep only unrelated individuals. We allowed for cryptic relatedness by using the mixed model in EPACTS for association testing (line 105). Similarly, in the replication analysis, we did not keep only unrelated individuals but rather we adjusted for relatedness (line 131).

- I suppose all the alleles shown in the manuscript tables/text are on the + strand? please specify.

Response: All alleles are reported with respect to the positive strand (lines 95-96).

- Replication studies: please clearly state the genome build used in methods and tables (same as discovery study so, hg19?).

Response: We used hg19 in all analyses (lines 95 and 133).

- All R packages: provide version.

Response: We provided version numbers for all R packages (lines 104, 153, and 169).

- Reference the R software and provide version.

Response: We have added a reference and a version (line 108).

Row 82: Is there a missing "." here ? if not, the sentence is not really clear.

Response: We added the missing punctuation (line 75).

Row 83: "Among Europeans" : please specify how the EUR ancestry was determined (self report, genetic data ? or both ?).

Response: We clarified the text (line 78).

Row 88: use "race and ethnicity" instead or "race" only.

Response: We have added ethnicity (line 82).

Row 89: T2D definitions in row 79 and 91 are slightly different, please clarify.

Response: We have clarified the definition (lines 71-73).

Row 96: GWAS QC: no sex check control performed? no ethnicity check? Also maybe provide some references for the QC of replication studies?

Response: For in-house data sets (AADM, HUFS, and CADM) for which we collected individual-level data from study participants, we performed sex checks (lines 127-128). For the data sets from dbGaP (ARIC, FHS, and MESA), we relied on documentation available within dbGaP (lines 128-129). We do not know what an ethnicity check is, so we are certain we did not perform one. No additional references are required.

Row 107: Can the author provide a brief justification as to why they used a two-stage mixed model? (i.e. first regressing on Age sex BMI on log FPG, then use ranked inverse normalized residuals for stage 2? Also, does this mean that all FPG related results (Beta, SE, etc) we see in the tables are back transformed and are in mmol/l? if yes, this should be mentioned.

Response: For discovery and replication GWAS, fasting glucose values were ranked and inverse normalized, because skewness resulting from the inclusion of untreated cases made the log transformation insufficient to achieve approximate normality (lines 102 and 130). As rank-based transformations cannot be meaningfully back transformed, we relied on the log transformation and back transformation for the estimation of βs and SEs in the original units of mmol/l (lines 165-166). Obtaining residuals in stage 1 is done for computational convenience, as the performance of additional analyses is easier with precomputed residuals.

Row 111: How many PCs adjusted for?

Response: In the discovery GWAS, we adjusted for the top 3 PCs (line 104). For each replication study, we also adjusted for the top three PCs (line 131).

Row 118: What was the random factor in the mixed models?

Response: There is no mention of random factors or mixed models in line 118. The random effect in the mixed model was the standard genetic relationship (kinship) matrix used in EPACTS (line 105).

Row 118-119: Please provide a full dbGAP accession number with the version (phsxxxxxx.vx.px format).

Response: The dbGaP accession numbers are provided in the Acknowledgements section (lines 354, 365, and 372).

Row 121: Add citation to the dbGAP database itself.

Response: We added the URL to dbGaP to the main text (line 71).

Row 126: Consider giving the number of case controls by ethnicity form each study in a supplementary table.

Response: We added this information to S1 Table (line 201).

Row 131: For each study (both stage 1 and replication studies) and for each ethnic group, provide a supplementary table with genotyping array, imputation panel used, filtering criteria (e.g. info score and MAF) and the resulting number of SNPs used (in the analysis in addition to the paragraph lines 131 to 138).

Response: We provided this information in S1 Table (line 201).

Row 164: Add ref to R package Meta.

Response: We added a citation for the package (line 169).

Row 152: How do you define the presence of "LD between SNP".

Response: For fine mapping, LD between SNPs is not defined in terms of presence. As is standard, we used a square matrix of pairwise covariance values.

Results:

Row 188: "The associations at all three lead SNPs were replicated", please specify that this is for the overall Meta-A, and briefly describe the replication results by ethnic groups.

Response: We clarified the text (line 214). Results by ethnic group are provided in S3 Table (line 214).

Row 188: "Of the two suggestive loci": Please specify what you mean by suggestive locus ( 5*10-7<P <5*10-8 ?)

Response: We clarified the text (line 215).

Row 192 + 198: SIFT, Polyphen-2 and CADD not mentioned at all in the methods. Please correct. also Ensembl is referenced instead of the programs and original papers themselves, if SIFT, Polyphen-2 and CADD results were extracted from Ensembl, then this should be specified in the methods, and references should include the original SIFT, Polyphen-2 and CADD papers in addition to Ensembl.

Response: The reviewer is correct that we accessed and retrieved information from Ensembl, and thus we cited Ensembl as appropriate. We did no formal analysis with SIFT, Polyphen-2, or CADD values, hence there are no methods or results to report. We discussed our results in the context of these preexisting data. Citing secondary references in this context is not the norm. However, we added descriptions and citations for SIFT, Polyphen-2, and CADD (lines 143-150).

row 193 + Figure S5: "The model protein structure of GCKR with L446P predicted an altered interaction between GCKR and GCK Figure S5)": not very clear from the figure S5 how the "interaction is altered", do you have a metric you use? a prediction P-value? also, maybe you could add arrows that point to where we're supposed to look on Figure S5 + specify what the purple color means?

Response: We redrew S6 Fig with the addition of labels. Also, we revised the text to indicate, as depicted in the figure, that L446 (yellow) and L446P (purple) are positioned differently (lines 227-228).

Row 230: eQTL and sQTL analysis not described in methods and results.

Response: There is no analysis to describe in either the Methods or Results sections. We simply retrieved and discussed existing information in the GTEx database. We provided eQTL and sQTL descriptions and accession information (lines 172-176 and 374-375). 

Row 209: Please correct "nomarl".

Response: We deleted the sentence.

Row 209: "compared with normal values are less ...". Please consider changing the phrasing.

Response: We deleted the sentence.

Row 210: Please correct "mmol/".

Response: We deleted the sentence.

Discussion:

Row 236: it is a bit confusing that authors describe rs560887 as lead SNP after fine mapping, having a low functional impact (row 195), but then discuss eQTL and sQTL effects in the discussion, and later suggest that rs573225 might be a better candidate. Was rs573225 included in fine mapping? what was its posterior probability? has the fine mapping performed in the different ethnic groups or restricted to EUR?

Response: It is correct that rs560087 was the lead SNP from the discovery analysis. The data are not conclusive whether rs560087, rs573225, both, or neither is/are likely causal. Regional fine mapping was performed using the discovery data (that is, restricted to EUR). The marginal posterior inclusion probability for rs560087 was 0.67 (S4 Fig). The lead SNP rs560087 is in strong LD with rs573225 in EUR (lines 287-288). The tagged marker rs573225 was included in the fine mapping analysis. The marginal posterior inclusion probability for rs573225 was 0.000404. The fine mapping analysis accounted for the absence or presence of promoter histone marks, enhancer histone marks, DNAse I hypersensitive sites, or bound proteins (lines 156-158). The fine mapping analysis did not account for eQTL annotation, sQTL annotation, predicted regulatory motifs, or CADD scores. Thus, we discuss these additional annotations. First, both rs560087 and rs573225 are annotated as both eQTL and sQTL (lines 281-284 and 290-292). Second, rs573225 had a higher CADD score than rs560087 (lines 292-293). Third, unlike rs560087, rs573225 was annotated with a predicted regulatory motif (lines 293-296). Therefore, there are two lines of evidence not accounted for by the CAVIARBF analysis suggesting that rs573225 might be a better candidate than rs560087 (lines 296-298).

Row 258-261: " based on the sequence to structure... for predicted protein function": should not be included in discussion, it reads more like a methods paragraph.

Response: We deleted the sentence.

Introduction:

Row 37: Please replace ".." by "."

Response: We fixed the punctuation (line 35).

Abstract:

Row 34: The authors should not just mention the results of 0 vs 6 T alleles and emphasize the gradual change of outcome based on the number of risk alleles (linear trend) instead.

Response: We revised the sentence to emphasize the additive nature of the genetic protection (lines 32-33).

Row 37: "Since non of the individuals homozygous for the ... identify individuals at low vs. high risk of developing type 2 diabetes": Not sure about this statement... I think if you derive a genetic score using more than 3 SNPs it will have a far better predictive power than a score with 3 SNPs.

Response: We are sure about our statement. The reviewer’s claim may very well be true, but given the data presented in this manuscript, the claim must be viewed as speculative and unsubstantiated, and as such, inappropriate for the Abstract.

All Tables, Figures and Supplementary Figures: Please add abbreviations and units when applicable.

Response: We have verified that all tables, figures, and supplementary files include applicable abbreviations, footnotes, and units.

Table S3:

-Add an Ethnicity column.

Response: We added the column.

-Show meta-analysis heterogeneity results.

Response: We added one column to present heterogeneity (Cochran’s Q value).

-Show meta-A results by ethnic group (Europeans, Africans + African descent, Chinese, Latinos)

Response: We added meta-analysis results by group.

Figure S3: Figure Resolution too low.

Response: We redrew the figure.

Figure S6: The figure title does not reflect the fact that you are testing the number of protective alleles.

Response: The meta-analysis is a priori agnostic as to direction of effect (i.e., the test is two-tailed). In fact, we are not testing the number of protective alleles.

Figure S7: The figure title does not reflect the fact that you are re testing the number of protective alleles.

Response: The meta-analysis is a priori agnostic as to direction of effect (i.e., the test is two-tailed). In fact, we are not testing the number of protective alleles.

References: I believe I found a couple of references that do not match what is being said in the main text, please double check the references.

Response: We do not know to which specific references the reviewer is referring. After adding the new references requested by the reviewer, we checked all references.

---

## [Decision Letter · Decision Letter 1]

20 May 2022

Additive genetic effect of GCKR, G6PC2, and SLC30A8 variants on fasting glucose levels and risk of type 2 diabetes

PONE-D-21-24235R1

Dear Dr. Chen,

We’re pleased to inform you that your manuscript has been judged scientifically suitable for publication and will be formally accepted for publication once it meets all outstanding technical requirements.

Kind regards,

Nicholette D. Palmer, Ph.D.

Academic Editor

PLOS ONE

Additional Editor Comments (optional):

Reviewers' comments:

Reviewer's Responses to Questions

**Comments to the Author**

1. If the authors have adequately addressed your comments raised in a previous round of review and you feel that this manuscript is now acceptable for publication, you may indicate that here to bypass the “Comments to the Author” section, enter your conflict of interest statement in the “Confidential to Editor” section, and submit your "Accept" recommendation.

Reviewer #1: All comments have been addressed

Reviewer #2: All comments have been addressed

2. Is the manuscript technically sound, and do the data support the conclusions?

Reviewer #1: Yes

Reviewer #2: Yes

3. Has the statistical analysis been performed appropriately and rigorously? 

Reviewer #1: Yes

Reviewer #2: Yes

4. Have the authors made all data underlying the findings in their manuscript fully available?

Reviewer #1: Yes

Reviewer #2: (No Response)

5. Is the manuscript presented in an intelligible fashion and written in standard English?

Reviewer #1: Yes

Reviewer #2: Yes

6. Review Comments to the Author

Reviewer #1: The authors addressed all my concerns. I have no further comments.

Reviewer #2: (No Response)

7. PLOS authors have the option to publish the peer review history of their article (what does this mean?). If published, this will include your full peer review and any attached files.

Reviewer #1: No

Reviewer #2: No

---

## [Editor Report · Acceptance letter]

25 May 2022

PONE-D-21-24235R1 

Additive genetic effect of *iGCKR, G6PC2,* and *SLC30A8* variants on fasting glucose levels and risk of type 2 diabetes 

Dear Dr. Chen:

I'm pleased to inform you that your manuscript has been deemed suitable for publication in PLOS ONE. Congratulations! Your manuscript is now with our production department. 

Kind regards, 

on behalf of

Dr. Nicholette D. Palmer 

Academic Editor

PLOS ONE